# Learning Additive Exponential Family Graphical Models via $\ell_{2,1}$-norm Regularized M-Estimation

**Xiao-Tong Yuan**[†]  **Ping Li**[‡§]  **Tong Zhang**[‡]  **Qingshan Liu**[†]  **Guangcan Liu**[†]

†B-DAT Lab, Nanjing University of Info. Sci.&Tech.
Nanjing, Jiangsu, 210044, China
‡Depart. of Statistics and §Depart. of Computer Science, Rutgers University
Piscataway, NJ, 08854, USA
{xtyuan,qsliu, gcliu}@nuist.edu.cn, {pingli,tzhang}@stat.rutgers.edu

## Abstract

We investigate a subclass of exponential family graphical models of which the sufficient statistics are defined by arbitrary additive forms. We propose two $\ell_{2,1}$-norm regularized maximum likelihood estimators to learn the model parameters from i.i.d. samples. The first one is a joint MLE estimator which estimates all the parameters simultaneously. The second one is a node-wise conditional MLE estimator which estimates the parameters for each node individually. For both estimators, statistical analysis shows that under mild conditions the extra flexibility gained by the additive exponential family models comes at almost no cost of statistical efficiency. A Monte-Carlo approximation method is developed to efficiently optimize the proposed estimators. The advantages of our estimators over Gaussian graphical models and Nonparanormal estimators are demonstrated on synthetic and real data sets.

## 1 Introduction

As an important class of statistical models for exploring the interrelationship among a large number of random variables, undirected graphical models (UGMs) have enjoyed popularity in a wide range of scientific and engineering domains, including statistical physics, computer vision, data mining, and computational biology. Let $X = [X_1, ..., X_p]^\top$ be a $p$-dimensional random vector with each variable $X_i$ taking values in a set $\mathcal{X}$. Suppose $G = (V, E)$ is an undirected graph consists of a set of vertices $V = \{1, ..., p\}$ and a set of unordered pairs $E$ representing edges between the vertices. The pairwise UGMs over $X$ corresponding to $G$ can be written as the following exponential family distribution:

$$\mathbb{P}(X; \theta) \propto \exp\left\{\sum_{s \in V} \theta_s \varphi_s(X_s) + \sum_{(s,t) \in E} \theta_{st} \phi_{st}(X_s, X_t)\right\}. \tag{1}$$

In such a pairwise model, $(X_s, X_t)$ are conditionally independent (given the rest of the variables) if and only if the weight $\theta_{st}$ is zero. The most popular instances of pairwise UGMs are Gaussian graphical models (GGMs) [19, 2] for real-valued random variables and Ising (or Potts) models [15] for binary or finite nominal discrete random variables. More broadly, in order to derive multivariate graphical models from univariate exponential family distributions (such as the Gaussian, binomial/multinomial, Poisson, exponential distributions, etc.), the exponential family graphical models (EFGMs) [27, 21] were proposed as a unified framework to learn UGMs with node-wise conditional distributions arising from generalized linear models (GLMs).

## 1.1 Overview of contribution

A fundamental issue that arises in UGMs is to specify sufficient statistics, i.e., $\{\varphi_s(X_s), \phi_{st}(X_s, X_t)\}$, for modeling the interactions among variables. It is noteworthy that most prior pairwise UGMs use pairwise product of variables (or properly transformed variables) as pairwise sufficient statistics [16, 11, 27]. This is clearly restrictive in modern data analysis tasks where the underlying pairwise interactions among variables are more often than not highly complex and unknown a prior. The goal of this work is to remove such a restriction and explore the feasibility (in theory and practice) of defining sufficient statistics in an additive formulation to approximate the underlying unknown sufficient statistics. To this end, we consider the following *Additive Exponential Family Graphical Model* (AdEFGM) distribution with joint density function:

$$\mathbb{P}(X; f) = \exp\left\{\sum_{s \in V} f_s(X_s) + \sum_{(s,t) \in E} f_{st}(X_s, X_t) - A(f)\right\}, \tag{2}$$

where $f_s : \mathcal{X} \to \mathbb{R}$ and $f_{st}(\cdot, \cdot) : \mathcal{X}^2 \to \mathbb{R}$ are respectively node-wise and pairwise statistics, and $A(f) := \log \int_{\mathcal{X}^p} \exp\left\{\sum_{s \in V} f_s(X_s) + \sum_{(s,t) \in E} f_{st}(X_s, X_t)\right\} dX$ is the log-partition function. We require the condition $A(f) < \infty$ holds so that the definition of probability is valid. In this paper, we assume the formulations of sufficient statistics $f_s$ and $f_{st}$ are *unknown* but they admit linear representations over two sets of pre-fixed basis functions $\{\varphi_k(\cdot), k = 1, 2, ..., q\}$ and $\{\phi_l(\cdot, \cdot), l = 1, 2, ..., r\}$, respectively. That is,

$$f_s(X_s) = \sum_{k=1}^q \theta_{s,k} \varphi_k(X_s), \quad f_{st}(X_s, X_t) = \sum_{l=1}^r \theta_{st,l} \phi_l(X_s, X_t), \tag{3}$$

where $q$ and $r$ are the truncation order parameters. In the formulation (3), the choice of basis and their sizes is flexible and task-dependent. For instance, if the mapping functions $f_s$ and $f_{st}$ are periodic, then we can choose $\{\varphi_k(\cdot)\}$ as 1-D Fourier basis and $\{\phi_l(\cdot, \cdot)\}$ as 2-D Fourier basis. As another instance, the basis $\{\phi_l\}$ can be chosen as multiple kernels which are commonly used in computer vision tasks. Specially, when $q = r = 1$, $\phi_l(X_s, X_t) = X_s X_t$ and $\varphi_k(X_s)$ is fixed as certain parametric function, AdEFGM reduces to the standard EFGM [27, 21]. In general cases, by imposing an additive structure on the sufficient statistics $f_s$ and $f_{st}$, AdEFGM is expected to be able to capture more complex interactions among variables beyond pairwise product.

As the core contribution of this paper, we propose two $\ell_{2,1}$-norm regularized maximum likelihood estimation (MLE) estimators to learn the weights of AdEFGM in high dimensional settings. The first estimator is formulated as an $\ell_{2,1}$-norm regularized MLE to jointly estimate all the parameters in the model. The second estimator is formulated as an $\ell_{2,1}$-norm regularized node-wise conditional MLE to estimate the parameters associated with each node individually. Theoretically, we prove that under mild conditions the joint MLE estimator achieves convergence rate $O((\sqrt{(2|E| + p) \ln p / n})$ where $|E|$ while the node-wise conditional estimator achieves convergence rate $O(\sqrt{(d + 1) \ln p / n})$ in which $d$ is the degree of the underlying graph $G$. Computationally, we propose a Monte-Carlo approximation scheme to efficiently optimize the estimators via proximal gradient descent methods. We conduct numerical studies on simulated and real data to support our claims. The simulation results confirm that, when the data are drawn from an underlying UGMs with highly nonlinear sufficient statistics, our estimators significantly outperform GGMs and Nonparanormal [10] estimators in most cases. The experimental results on a stock price data show that our estimators are able to recover more accurate category links among stocks than GMMs and Nonparanormal estimators.

## 1.2 Related work

In order to model random variables beyond parametric UGMs such as GGMs and Ising models, researchers recently investigated semi-parametric/nonparametric extensions of these parametric models. The Nonparanormal [11] and copula-based methods [5] are semi-parametric graphical models which assume that data is Gaussian after applying a monotone transformation. More broadly, one could learn transformations of the variables and then fit any parametric UGMs (like EFGMs) over the transformed variables. In [10, 26], two rank-based estimators were used to estimate correlation matrix and then fit the GGMs. In [24], a semi-parametric method was proposed to fit the conditional

means of the features with an arbitrary additive formulation. The Semi-EFGM proposed in [28] is a semi-parametric rank-based conditional estimator for exponential family graphical models. In [1], a kernel method was proposed for learning the structure of graphical models by treating variables as Gaussians in a mapped high-dimensional feature space. In [7], Gu proposed a functional minimization framework to estimate the nonparametric model (1) over a Reproducing Hilbert Kernel Space (RKHS). Nonparametric exponential family graphical models based on score matching loss were investigated in [9, 20]. The forest density estimation [8] is a fully nonparametric method for estimating UGMs with structure restricted to be a forest. In contrast to all these existing semi-parametric/nonparametric models, our approach is novel in model definition and computation: we impose a simple additive structure on sufficient statistics to describe complex interactions between variables and use Monte-Carlo approximation to estimate the intractable normalization constant for efficient optimization.

## 1.3 Notation and organization

**Notation** Let $\theta = \{\theta_{s,k}, \theta_{st,l} : s \in V, k = 1, 2, ..., (s,t) \in V^2, s \neq t, l = 1, 2, ...\}$ be a vector of parameters associated with AdEFGM and $\mathcal{G} = \{\{(s,k)\}_k, \{(st,l)\}_l : s \in V, (s,t) \in V^2, s \neq t\}$ be a group induced by the additive structures of nodes and edges. We conventionally define the following grouped-norm related notations: $\|\theta\|_{2,1} = \sum_{g \in \mathcal{G}} \|\theta_g\|$, $\|\theta\|_{2,\infty} = \max_{g \in \mathcal{G}} \|\theta_g\|$, $\mathrm{supp}(\theta, \mathcal{G}) = \{g \in \mathcal{G} : \|\theta_g\| \neq 0\}$ and $\|\theta\|_{2,0} = |\mathrm{supp}(\theta, \mathcal{G})|$. For any $S \subseteq \mathcal{G}$, these notations can be defined restrictively over $\theta_S$. We denote $\bar{S} = \mathcal{G} \setminus S$ the complement of $S$ in $\mathcal{G}$.

**Organization.** The remaining of this paper is organized as follows: In §2, we present two maximum likelihood estimators for learning the model parameters of AdEFGM. The statistical guarantees of the proposed estimators are analyzed in §3. Monte-Carlo simulations and experimental results on real stock price data are presented in §4. Finally, we conclude this paper in §5. Due to space limit, all the technical proofs of theoretical results are deferred to an appendix section which is included in the supplementary material.

## 2 $\ell_{2,1}$-norm Regularized MLE for AdEFGM

In this section, we investigate the problem of estimating the parameters of AdEFGM in high dimensional settings. By substituting (3) into (2), the distribution of an AdEFGM can be converted to the following form:
$$\mathbb{P}(X; \theta) = \exp\{B(X; \theta) - A(\theta)\}, \tag{4}$$
where $\theta = \{\theta_{s,k}, \theta_{st,l}\}$, and
$$B(X; \theta) := \sum_{s \in V, k} \theta_{s,k} \varphi_k(X_s) + \sum_{(s,t) \in E, l} \theta_{st,l} \phi_l(X_s, X_t), \quad A(\theta) := \log \int_{\mathcal{X}^p} \exp\{B(X; \theta)\}\, dX.$$

Suppose we have $n$ i.i.d. samples $\mathbb{X}_n = \{X^{(i)}\}_{i=1}^n$ drawn from the following AdEFGM with true parameters $\theta^*$:
$$\mathbb{P}(X; \theta^*) = \exp\{B(X; \theta^*) - A(\theta^*)\}. \tag{5}$$
An important goal of graphical model learning is to estimate the true parameters $\theta^*$ from the observed data $\mathbb{X}_n$. The more accurate parameter estimation is, the more accurate we are able to recover the underlying true graph structure. We next propose two $\ell_{2,1}$-norm regularized maximum likelihood estimation (MLE) methods for joint and node-conditional learning of parameters, respectively.

## 2.1 Joint MLE estimation

Given the sample set $\mathbb{X}_n = \{X^{(i)}\}_{i=1}^n$, the negative log-likelihood of the joint distribution (5) is:
$$L(\theta; \mathbb{X}_n) = -\frac{1}{n} \sum_{i=1}^n B(X^{(i)}; \theta) + A(\theta).$$

It is trivial to verify $L(\theta; \mathbb{X}_n)$ has the following first order derivative (see, e.g., [25]):
$$\frac{\partial L}{\partial \theta_{s,k}} = \mathbb{E}_\theta[\varphi_k(X_s)] - \frac{1}{n} \sum_{i=1}^n \varphi_k(X_s^{(i)}), \quad \frac{\partial L}{\partial \theta_{st,l}} = \mathbb{E}_\theta[\phi_l(X_s, X_t)] - \frac{1}{n} \sum_{i=1}^n \phi_l(X_s^{(i)}, X_t^{(i)}), \tag{6}$$

where the expectation $\mathbb{E}_\theta[\cdot]$ is taken over the joint distribution (2). Also, it is well known that $L(\theta; \mathbb{X}_n)$ is convex in $\theta$.

In order to estimate the parameters which are expected to be sparse in edge level due to the potential sparse structure of graph, we consider the following $\ell_{2,1}$-norm regularized MLE estimator:

$$\hat{\theta}_n = \arg\min_\theta \left\{ L(\theta; \mathbb{X}_n) + \lambda_n \|\theta\|_{2,1} \right\}, \tag{7}$$

where $\|\theta\|_{2,1} = \sum_{s \in V} \left( \sum_{k=1}^q \theta_{s,k}^2 \right)^{1/2} + \sum_{(s,t) \in V^2, s \neq t} \left( \sum_{l=1}^r \theta_{st,l}^2 \right)^{1/2}$ is the $\ell_{2,1}$-norm with respect to the basis statistics and $\lambda_n > 0$ is the regularization strength parameter dependent on $n$. The $\ell_{2,1}$-norm penalty is used to promote edgewise sparsity as the graph structure is expected to be sparse in high dimensional settings.

## 2.2 Node-conditional MLE estimation

Recent state of the art methods for learning UGMs suggest a natural procedure deriving multivariate graphical models from univariate distributions [12, 15, 27]. The common idea in these methods is to learn the graph structure by estimating node-neighborhoods, or by fitting the node-conditional distribution of each individual node conditioned on the rest of the nodes. Indeed, these node-wise fitting methods have been shown to have strong statistical guarantees and attractive computational performance. Inspired by these approaches, we propose an alternative estimator to estimate the weights of sufficient statistics associated with each individual node. With a slight abuse of notation, we denote $\theta_{\mathbf{s}}$ a subvector of $\theta$ associated with node $s$, i.e.,

$$\theta_{\mathbf{s}} := \{\theta_{s,k} \mid k = 1, ..., q\} \cup \{\theta_{st,l} \mid t \in N(s), l = 1, ..., r\},$$

where $N(s)$ is the neighborhood of $s$. Given the joint distribution (4), it is easy to show that the conditional distribution of $X_s$ given the rest variables, $X_{\backslash s}$, is written by:

$$\mathbb{P}(X_s \mid X_{\backslash s}; \theta_{\mathbf{s}}) = \exp\left\{ C(X_s \mid X_{\backslash s}; \theta_{\mathbf{s}}) - D(X_{\backslash s}; \theta_{\mathbf{s}}) \right\}, \tag{8}$$

where $C(X_s \mid X_{\backslash s}; \theta_{\mathbf{s}}) := \sum_k \theta_{s,k} \varphi_k(X_s) + \sum_{t \in N(s),l} \theta_{st,l} \phi_l(X_s, X_t)$, and $D(X_{\backslash s}; \theta_{\mathbf{s}}) := \log \int_{\mathcal{X}} \exp\left\{ C(X_s \mid X_{\backslash s}; \theta_{\mathbf{s}}) \right\} dX_s$ is the log-partition function which ensures normalization. We note that the condition $A(\theta) < \infty$ for the joint log-partition function implies $D(X_{\backslash s}; \theta_{\mathbf{s}}) < \infty$.

In order to estimate the parameters associated with a node, we consider using the sparsity regularized conditional maximum likelihood estimation. Given $n$ independent samples $\mathbb{X}_n$ drawn from (5), we can write the negative log-likelihood of the conditional distribution as:

$$\tilde{L}(\theta_{\mathbf{s}}; \mathbb{X}_n) = \frac{1}{n} \sum_{i=1}^n \left\{ -C(X_s^{(i)} \mid X_{\backslash s}^{(i)}; \theta_{\mathbf{s}}) + D(X_{\backslash s}^{(i)}; \theta_{\mathbf{s}}) \right\}.$$

It is standard that $\tilde{L}(\theta_{\mathbf{s}}; \mathbb{X}_n)$ is convex with respect to $\theta_{\mathbf{s}}$ and it has the following first-order derivative:

$$\begin{aligned}
\frac{\partial \tilde{L}(\theta_{\mathbf{s}}; \mathbb{X}_n)}{\partial \theta_{s,k}} &= \frac{1}{n} \sum_{i=1}^n \left\{ -\varphi_k(X_s^{(i)}) + \mathbb{E}_{\theta_{\mathbf{s}}}[\varphi_k(X_s) \mid X_{\backslash s}^{(i)}] \right\}, \\
\frac{\partial \tilde{L}(\theta_{\mathbf{s}}; \mathbb{X}_n)}{\partial \theta_{st,l}} &= \frac{1}{n} \sum_{i=1}^n \left\{ -\phi_l(X_s^{(i)}, X_t^{(i)}) + \mathbb{E}_{\theta_{\mathbf{s}}}[\phi_l(X_s, X_t^{(i)}) \mid X_{\backslash s}^{(i)}] \right\},
\end{aligned} \tag{9}$$

where the expectation $\mathbb{E}_{\theta_{\mathbf{s}}}[\cdot \mid X_{\backslash s}]$ is taken over the node-wise conditional distribution (8).

Let us consider the following $\ell_{2,1}$-norm regularized conditional MLE formulation associated with the variable $X_s$:

$$\hat{\theta}_{\mathbf{s}}^n = \arg\min_{\theta_{\mathbf{s}}} \left\{ \tilde{L}(\theta_{\mathbf{s}}; \mathbb{X}_n) + \lambda_n \|\theta_{\mathbf{s}}\|_{2,1} \right\}, \tag{10}$$

where $\|\theta_{\mathbf{s}}\|_{2,1} = \left( \sum_{k=1}^q \theta_{s,k}^2 \right)^{1/2} + \sum_{t \neq s} \left( \sum_{l=1}^r \theta_{st,l}^2 \right)^{1/2}$ is the grouped $\ell_{2,1}$-norm with respect to the node-wise and pairwise basis associated with $s$ and $\lambda_n > 0$ controls the regularization strength.

## 2.3 Computation via Monte-Carlo approximation

We consider using proximal gradient descent methods [22] to solve the composite optimization problems in (7) and (10). For both estimators, the major computational overhead is to iteratively calculate the expectation terms involved in the gradients $\nabla L(\theta; \mathbb{X}_n)$ and $\nabla \tilde{L}(\theta_\mathbf{s}; \mathbb{X}_n)$. In general, these expectation terms have no close-form for exact calculation and sampling methods such as importance sampling and MCMC are usually needed for approximate estimation. There are, however, two challenging issues with such a sampling based optimization procedure: (1) the multivariate sampling methods typically suffer from high computational cost even when the dimensionality $p$ is moderately large; and (2) the non-vanishing sampling error of gradient will accumulate during the iteration which according to the results in [18] will deteriorate the overall convergence performance. Obviously, the main source of these challenges is caused by the intractable log-partition terms appeared in the estimators.

To more efficiently apply the first-order methods without suffering from iterative sampling and error accumulation, it is a natural idea to replace the log-partition terms by a Monte-Carlo approximation and minimize the resulting approximated formulation. Taking the joint estimator (7) as an example, we resort to the basic importance sampling method to approximately estimate the log-partition term $A(\theta) = \log \int_{\mathcal{X}^p} \exp \{B(X; \theta)\} \, dX$. Assume we have $m$ i.i.d. samples $\mathbb{Y}_m = \{Y^{(j)}\}_{j=1}^m$ drawn from a random vector $Y \in \mathcal{X}^p$ with known probability density $\mathbb{P}(Y)$. Given $\theta$, an importance sampling estimate of $\exp\{A(\theta)\}$ is given by

$$\exp\{\hat{A}(\theta; \mathbb{Y}_m)\} = \frac{1}{m} \sum_{j=1}^m \frac{\exp \left\{ B(Y^{(j)}; \theta) \right\}}{\mathbb{P}(Y^{(j)})}.$$

We consider the following Monte-Carlo approximation to the estimator (7):

$$\hat{\hat{\theta}}_n = \arg \min_\theta \left\{ \hat{L}(\theta; \mathbb{X}_n, \mathbb{Y}_m) + \lambda_n \|\theta\|_{2,1} \right\}, \tag{11}$$

where $\hat{L}(\theta; \mathbb{X}_n, \mathbb{Y}_m) = -\frac{1}{n} \sum_{i=1}^n B(X^{(i)}; \theta) + \hat{A}(\theta; \mathbb{Y}_m)$. Since the random samples $\mathbb{Y}_m$ are fixed in (11), the sampling operation can be avoided in the computation of $\nabla \hat{L}(\theta; \mathbb{X}_n, \mathbb{Y}_m)$. Concerning the accuracy of the approximate estimator (11), the following result guarantees that, with high probability, the minimizer of the approximate estimator (11) is suboptimal to the population estimator (7) with suboptimality $\mathcal{O}(1/\sqrt{m})$. A proof of this proposition is provided in A.1 (see the supplementary material).

**Proposition 1.** *Assume that $\mathbb{P}(Y) > 0$. Then the following inequality holds with high probability:*

$$L(\hat{\hat{\theta}}_n; \mathbb{X}_n) + \lambda_n \|\hat{\hat{\theta}}_n\|_{2,1} \leq L(\hat{\theta}_n; \mathbb{X}_n) + \lambda_n \|\hat{\theta}_n\|_{2,1} + \frac{2.58\hat{\sigma} \left( \exp\{-A(\hat{\hat{\theta}}_n\} + \exp\{-\hat{A}(\hat{\theta}_n; \mathbb{Y}_m)\} \right)}{\sqrt{m}},$$

*where $\hat{\sigma}_n = \frac{1}{m} \sum_{j=1}^m \left( \frac{\exp\{B(Y^{(j)}; \hat{\theta}_n)\}}{\mathbb{P}(Y^{(j)})} - \exp\{\hat{A}(\hat{\theta}_n; \mathbb{Y}_m)\} \right)^2.$*

A similar Monte-Carlo approximation strategy can be applied to the node-wise MLE estimator (10).

## 3 Statistical Analysis

In this section, we provide statistical guarantees on parameter estimation error for the joint MLE estimator (7) and the node-conditional estimator (10). In large picture, our analysis follows the techniques presented in [13, 30] by specifying the conditions under which these techniques can be applied to our setting.

### 3.1 Analysis of the joint estimator

We are interested in the concentration bounds of the random variables defined by

$$Z_{s,k} := \varphi_k(X_s) - \mathbb{E}_{\theta^*}[\varphi_k(X_s)], \quad Z_{st,l} := \phi_l(X_s, X_t) - \mathbb{E}_{\theta^*}[\phi_l(X_s, X_t)],$$

where the expectation $\mathbb{E}_{\theta^*}[\cdot]$ is taken over the underlying true distribution (5). By the "law of the unconscious statistician" we have $\mathbb{E}[Z_{s,k}] = \mathbb{E}[Z_{st,l}] = 0$. That is, $\{Z_{s,k}\}$ and $\{Z_{st,l}\}$ are zero-mean random variables. We introduce the following technical condition on $\{Z_{s,k}, Z_{st,l}\}$ which we will show to guarantee the gradient $\nabla L(\theta^*; \mathbb{X}_n)$ vanishes exponentially fast, with high probability, as sample size increases.

**Assumption 1.** *For all $(s, k)$ and all $(s, t, l)$, we assume that there exist constants $\sigma > 0$ and $\zeta > 0$ such that for all $|\eta| \leq \zeta$,*

$$\mathbb{E}[\exp\{\eta Z_{s,k}\}] \leq \exp\left\{\sigma^2 \eta^2 / 2\right\}, \quad \mathbb{E}[\exp\{\eta Z_{st,l}\}] \leq \exp\left\{\sigma^2 \eta^2 / 2\right\}.$$

This assumption essentially imposes an exponential-type bound on the moment generating function of the random variables $Z_{s,k}, Z_{st,l}$.

It is well known that the Hessian $\nabla^2 L(\theta; \mathbb{X}_n)$ is positive semidefinite at any $\theta$ and it is independent on the sample set $\mathbb{X}_n$. We also need the following condition which guarantees the restricted positive definiteness of $\nabla^2 L(\theta; \mathbb{X}_n)$ over certain low dimensional subspace when $\theta$ is in the vicinity of $\theta^*$.

**Assumption 2** (**Locally Restricted Positive Definite Hessian**). *Let $S = supp(\theta^*; \mathcal{G})$. There exist constants $\delta > 0$ and $\beta > 0$ such that for any $\theta \in \{\|\theta - \theta^*\| \leq \delta\}$, the inequality $\vartheta^\top \nabla^2 L(\theta; \mathbb{X}_n)\vartheta \geq \beta\|\vartheta\|^2$ holds for any $\vartheta \in \mathcal{C}_S := \{\|\theta_{\bar{S}}\|_{2,1} \leq 3\|\theta_S\|_{2,1}\}$.*

Assumption 2 requires that the Hessian $\nabla^2 L(\theta; \mathbb{X}_n)$ is positive definite in the cone $\mathcal{C}_S$ when $\theta$ lies in a local ball centered at $\theta^*$. This condition is a specification of the concept *restricted strong convexity* [30] to AdEFGM.

**Remark 1** (Minimal Representation). *We say an AdEFGM has* minimal representation *if there is a unique parameter vector $\theta$ associated with the distribution* (4). *This condition equivalently requires that there exists no non-zero $\vartheta$ such that $B(X; \vartheta)$ is equal to an absolute constant. This implies that for any $\theta$ and for all non-zero $\vartheta$,*

$$Var_\theta[B(X; \vartheta)] = \vartheta^\top \nabla^2 L(\theta; \mathbb{X}_n)\vartheta > 0.$$

*If AdEFGM has minimal representation at $\theta^*$, then there must exist sufficiently small constants $\delta > 0$ and $\beta > 0$ such that for any $\theta \in \{\|\theta - \theta^*\| \leq \delta\}$, $\vartheta^\top \nabla^2 L(\theta; \mathbb{X}_n)\vartheta \geq \beta\|\vartheta\|^2$. Therefore, Assumption 2 holds true when AdEFGM has minimal representation at $\theta^*$.*

The following theorem is our main result on the estimation error of the joint MLE estimator (7). A proof of this result is provided in Appendix A.2 in the supplementary material.

**Theorem 1.** *Assume that the conditions in Assumption 1 and Assumption 2 hold. If sample size $n$ satisfies*

$$n > \max\left\{\frac{6\max\{q, r\}\ln p}{\sigma^2 \zeta^2}, \frac{54c_0^2 \sigma^2 \max\{q, r\}\|\theta^*\|_{2,0}\ln p}{\delta^2 \beta^2}\right\},$$

*then with probability at least $1 - 2\max\{q, r\}p^{-1}$, the following inequality holds:*

$$\|\hat{\theta}_n - \theta^*\| \leq 3c_0 \beta^{-1} \sigma \sqrt{6\max\{q, r\}\|\theta^*\|_{2,0}\ln p/n}.$$

**Remark 2.** *The main message Theorem 1 conveys is that when $n$ is sufficiently large, the estimation error $\|\hat{\theta}_n - \theta^*\|$ vanishes at the order of $O(\sqrt{\max\{q, r\}(2|E| + p)\ln p/n})$ with high probability. This convergence rate matches the results obtained in [17, 16] for GGMs and the results in [10, 26] for Nonparanormal.*

## 3.2 Analysis of the node-conditional estimator

For the node-conditional estimator (10), we study the rate of convergence of the parameter estimation error $\|\hat{\theta}_{\mathbf{s}}^n - \theta_{\mathbf{s}}^*\|$ as a function of sample size $n$. We need Assumption 1 and the following assumption in our analysis.

**Assumption 3.** *For any node $s$, let $S = supp(\theta_{\mathbf{s}}^*; \mathcal{G})$. There exist constants $\tilde{\delta} > 0$ and $\tilde{\beta} > 0$ such that for any $\theta_{\mathbf{s}} \in \{\|\theta_{\mathbf{s}} - \theta_{\mathbf{s}}^*\| < \tilde{\delta}\}$, the inequality $\vartheta_{\mathbf{s}}^\top \nabla^2 \tilde{L}(\theta_{\mathbf{s}}; \mathbb{X}_n)\vartheta_{\mathbf{s}} \geq \tilde{\beta}\|\vartheta_{\mathbf{s}}\|^2$ holds for any $\vartheta_{\mathbf{s}} \in \tilde{\mathcal{C}}_S := \{\|(\theta_{\mathbf{s}})_{\bar{S}}\|_{2,1} \leq 3\|(\theta_{\mathbf{s}})_S\|_{2,1}\}$.*

The following is our main result on the convergence rate of node-conditional estimation error $\|\hat{\theta}_{\mathbf{s}}^n - \theta_{\mathbf{s}}^*\|$. A proof of this result is provided in Appendix A.3 in the supplementary material.

**Theorem 2.** *Assume that the conditions in Assumption 1 and Assumption 3 hold. If sample size $n$ satisfies*

$$n > \max\left\{\frac{6\max\{q,r\}\ln p}{\sigma^2\zeta^2}, \frac{216\tilde{c}_0^2\tilde{\sigma}^2\max\{q,r\}\|\theta_{\mathbf{s}}^*\|_{2,0}\ln p}{\delta^2\tilde{\beta}^2}\right\},$$

*then with probability at least $1 - 4\max\{q,r\}p^{-2}$, the following inequality holds:*

$$\|\hat{\theta}_{\mathbf{s}}^n - \theta_{\mathbf{s}}^*\| \leq 6\tilde{c}_0\tilde{\beta}^{-1}\sigma\sqrt{6\max\{q,r\}\|\theta_{\mathbf{s}}^*\|_{2,0}\ln p/n}.$$

**Remark 3.** *Theorem 2 indicates that with overwhelming probability, the estimation error $\|\hat{\theta}_{\mathbf{s}}^n - \theta_{\mathbf{s}}^*\| = O(\sqrt{(d+1)\ln p)/n})$ where $d$ is the degree of the underlying graph, i.e., $d = \max_{s \in V}\|\theta_{\mathbf{s}}^*\|_{2,0} - 1$. We may combine the parameter estimation errors from all the nodes as a global measurement of accuracy. Indeed, by Theorem 2 and union of probability we get that $\max_{s \in V}\|\hat{\theta}_{\mathbf{s}}^n - \theta_{\mathbf{s}}^*\| = O(\sqrt{(d+1)\ln p/n})$ holds with probability at least $1 - 4\max\{q,r\}p^{-1}$. This estimation error bound matches those for GGMs with neighborhood-selection-type estimators [29].*

# 4 Experiments

This section is devoted to showing the actual learning performance of AdEFGM. We first investigate graph structure recovery accuracy using simulation data (for which we know the ground truth), and then we apply our method to a stock price data for inferring the statistical dependency among stocks.

## 4.1 Monte-Carlo simulation

This is a proof-of-concept experiment. The purpose is to confirm that when the pairwise interactions of the underlying graphical models are highly nonlinear and unknown a prior, our additive estimator will be significantly superior to existing parametric/semi-parametric graphical models for inferring the structure of graphs. The numerical results of AdEFGM reported in this experiment are obtained by the joint MLE estimator in (7).

**Simulated data** Our simulation study employs a graphical model of which the edges are generated independently with probability $P$. We will consider the model under different levels of sparsity by adjusting the probability $P$. For simplicity purpose, we assume $f_s(X_s) \equiv 1$ and consider a nonlinear pairwise interaction function $f_{st}(X_s, X_t) = \cos(\pi(X_s - X_t)/5)$. We fit the data to the additive model (4) with a 2-D Fourier basis of size 8. Using Gibbs sampling, we generate a training sample of size $n$ from the true graphical model, and an independent sample of the same size from the same distribution for tuning the strength parameter $\lambda_n$. We compare performance for $n = 200$, varying values of $p \in \{50, 100, 150, 200, 250, 300\}$, and different sparsity levels under $P = \{0.02, 0.05, 0.1\}$, replicated 10 times for each configuration.

**Baselines** We compare the performance of our estimator to Graphical Lasso [6] as a GGM estimator and SKEPTIC [10] as a Nonparanormal estimator. In our implementation, we use a version of SKEPTIC with Kendall's tau to infer the correlation.

**Evaluation metric** To evaluate the support recovery performance, we use the standard F-score from the information retrieval literature. The larger F-score is, the better the support recovery performance. The numerical values over $10^{-3}$ in magnitude are considered to be nonzero.

**Results** Figure 1 shows the support recovery F-scores of the considered methods on the synthetic data. From this group of results we can observe that by using 2-D Fourier basis to approximate the unknown cosine distance function, AdEFGM is able to more accurately recover the underlying graph structure than the other two considered methods. The advantage of AdEFGM illustrated here is as expected because it is designed to automatically learn the unknown complex pairwise interactions while GGM and Nonparanormal are restrictive to certain UGMs with known sufficient statistics.

## 4.2 Stock price data

We further study the performance of AdEFGM on a stock price data. This data contains the historical prices of S&P500 stocks over 5 years, from January 1, 2008 to January 1, 2013. By taking out the

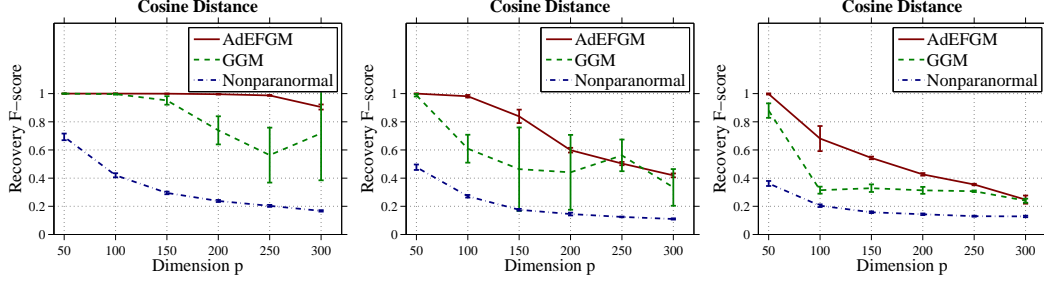

Figure 1: Simulated data: Support recovery F-score curves. Left panels: $P = 0.02$, Middle panels: $P = 0.05$, Right panels: $P = 0.1$.

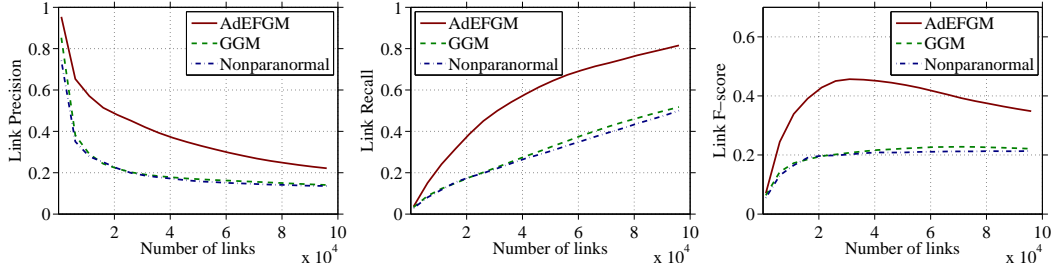

Figure 2: Stock price data S&P500: Category link precision, recall and F-score curves.

stocks with less than 5 years of history, we end up with 465 stocks, each having daily closing prices over 1,260 trading days. The prices are first adjusted for dividends and splits and the used to calculate daily log returns. Each day's return can be represented as a point in $\mathbb{R}^{465}$. To apply AdEFGM to this data, we consider the general model (4) with the 2-D Fourier basis being used to approximate the pairwise interaction between stocks $X_s$ and $X_t$. Since the category information of S&P500 is available, we measure the performance by Precision, Recall and F-score of the top $k$ links (edges) on the constructed graph. A link is regarded as *true* if and only if it connects two nodes belonging to the same category. We use the joint MLE estimator for this experiment. Figure 2 shows the curves of precision, recall and F-score as functions of $k$ in a wide range $[10^3, 10^5]$. It is apparent that AdEFGM significantly outperforms GGM and Nonparanormal for identifying correct category links. This result suggests that the interactions among the S&P500 stocks are highly nonlinear.

## 5 Conclusions

In this paper, we proposed and analyzed AdEFGMs as a generic class of additive undirected graphical models. By expressing node-wise and pairwise sufficient statistics as linear representations over a set of basis statistics, AdEFGM is able to capture complex interactions among variables which are not uncommon in modern engineering applications. We investigated two types of $\ell_{2,1}$-norm regularized MLE estimators for joint and node-conditional high dimensional estimation. Based on our theoretical justification and empirical observation, we can draw the following two conclusions: 1) the $\ell_{2,1}$-norm regularized AdEFGM learning is a powerful tool for inferring pairwise exponential family graphical models with unknown arbitrary sufficient statistics; and 2) the extra flexibility gained by AdEFGM comes at almost no cost of statistical and computational efficiency.

## Acknowledgments

Xiao-Tong Yuan and Ping Li were partially supported by NSF-Bigdata-1419210, NSF-III-1360971, ONR-N00014-13-1-0764, and AFOSR-FA9550-13-1-0137. Xiao-Tong Yuan is also partially supported by NSFC-61402232, NSFC-61522308, and NSFJP-BK20141003. Tong Zhang is supported by NSF-IIS-1407939 and NSF-IIS-1250985. Qingshan Liu is supported by NSFC-61532009. Guangcan Liu is supported by NSFC-61622305, NSFC-61502238 and NSFJP-BK20160040.

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
