[Supplementary Material]

# Abstract

This supplementary document contains all the technical proofs and several additional remarks for the NIPS'16 paper entitled "Learning Additive Exponential Family Graphical Models via $\ell_{2,1}$-norm Regularized M-Estimation". It is indeed the appendix section of the paper. The technical proofs are provided in Appendix A. The additional remarks on key assumptions are presented in Appendix B.

# A   Technical Proofs

## A.1   Proof of Proposition 1

*Proof.* Since $\mathbb{P}(Y) > 0$, it is standard to know (see, e.g., [14]) that an approximate $0.99$ confidence interval for $\exp\{A(\hat{\theta}_n)\}$ is $\exp\{\hat{A}(\hat{\theta}_n; \mathbb{Y}_m)\} \pm 2.58\hat{\sigma}_n/\sqrt{m}$ with $\hat{\sigma}$ given in the proposition. From the convexity of logarithm function we have the following inequality holds with confidence approximately $0.99$:

$$
\begin{aligned}
\hat{A}(\hat{\theta}_n; \mathbb{Y}_m) &= \log\left(\exp\{\hat{A}(\hat{\theta}_n; \mathbb{Y}_m)\}\right) \\
&\leq \log\left(\exp\{A(\hat{\theta}_n)\}\right) + \frac{2.58\hat{\sigma}\exp\{-\hat{A}(\hat{\theta}_n; \mathbb{Y}_m)\}}{\sqrt{m}} \\
&= A(\hat{\theta}_n) + \frac{2.58\hat{\sigma}\exp\{-\hat{A}(\hat{\theta}_n; \mathbb{Y}_m)\}}{\sqrt{m}}.
\end{aligned}
$$

Similarly, for $\hat{\hat{\theta}}_n$, we have the following inequality holds with confidence approximately $0.99$:

$$
A(\hat{\hat{\theta}}_n) \leq \hat{A}(\hat{\hat{\theta}}_n; \mathbb{Y}_m) + \frac{2.58\hat{\sigma}\exp\{-A(\hat{\hat{\theta}}_n)\}}{\sqrt{m}},
$$

From the preceding two inequalities and the optimality of $\hat{\hat{\theta}}_n$ we have that

$$
\begin{aligned}
L(\hat{\hat{\theta}}_n; \mathbb{X}_n) + \lambda_n\|\hat{\hat{\theta}}_n\|_{2,1} &\leq \hat{L}(\hat{\hat{\theta}}_n; \mathbb{X}_n, \mathbb{Y}_m) + \lambda_n\|\hat{\hat{\theta}}_n\|_{2,1} + \frac{2.58\hat{\sigma}\exp\{-A(\hat{\hat{\theta}}_n\}}{\sqrt{m}} \\
&\leq \hat{L}(\hat{\theta}_n; \mathbb{X}_n, \mathbb{Y}_m) + \lambda_n\|\hat{\theta}_n\|_{2,1} + \frac{2.58\hat{\sigma}\exp\{-A(\hat{\hat{\theta}}_n\}}{\sqrt{m}} \\
&\leq L(\hat{\theta}_n; \mathbb{X}_n) + \lambda_n\|\hat{\theta}_n\|_{2,1} + \frac{2.58\hat{\sigma}\left(\exp\{-A(\hat{\hat{\theta}}_n\} + \exp\{-\hat{A}(\hat{\theta}_n; \mathbb{Y}_m)\}\right)}{\sqrt{m}}
\end{aligned}
$$

holds with high probability. $\qquad\square$

## A.2   Proof of Theorem 1

We need the following result which indicates that under Assumption 1, $\{Z_{s,k}, Z_{st,l}\}$ satisfy a large deviation inequality.

**Lemma 1.** *If Assumption 1 holds, then for all $(s, k)$ and $(s, t, l)$ and any $\varepsilon \leq \zeta\sigma^2$ we have*

$$
\mathbb{P}\left(\left|\frac{1}{n}\sum_{i=1}^n \varphi_k(X_s^{(i)}) - \mathbb{E}_{\theta^*}[\varphi_k(X_s)]\right| > \varepsilon\right) \leq 2\exp\left\{-\frac{n\varepsilon^2}{2\sigma^2}\right\},
$$

$$
\mathbb{P}\left(\left|\frac{1}{n}\sum_{i=1}^n \phi_l(X_s^{(i)}, X_t^{(i)}) - \mathbb{E}_{\theta^*}[\phi_l(X_s, X_t)]\right| > \varepsilon\right) \leq 2\exp\left\{-\frac{n\varepsilon^2}{2\sigma^2}\right\}.
$$

*Proof.* From the definition and the "law of the unconscious statistician" we know that Assumption 1 identically requires

$$\mathbb{E}_{\theta^*}[\exp\{\eta(Z_{s,k})\}] \leq \exp\left\{\sigma^2\eta^2/2\right\}, \quad \mathbb{E}_{\theta^*}[\exp\{\eta(Z_{st,l})\}] \leq \exp\left\{\sigma^2\eta^2/2\right\}.$$

Since $X^{(i)}$ are i.i.d. samples of $X$, we have that $Z_{s,k}^{(i)} = \varphi_k(X_s^{(i)}) - \mathbb{E}_{\theta^*}[\varphi_k(X_s)]$ are also i.i.d. samples of $Z_{s,k}$. We use the exponential Markov inequality for the sum $Z = \sum_{i=1}^n Z_{s,k}^{(i)}$ and with a parameter $\eta > 0$:

$$\mathbb{P}\left(Z > \epsilon\right) = \mathbb{P}\left(\exp\{\eta Z\} > \exp\{\eta\epsilon\}\right) \leq \frac{\mathbb{E}[\exp\{\eta Z\}]}{\exp\{\eta\epsilon\}} = \frac{\prod_{i=1}^n \mathbb{E}\left[\exp\left\{\eta Z_{st}^{(i)}\right\}\right]}{\exp\{\eta\epsilon\}}.$$

If $\eta \leq \zeta$, Assumption 1 yields

$$\mathbb{P}\left(Z > n\varepsilon\right) \leq \frac{\exp\left\{n\sigma^2\eta^2/2\right\}}{\exp\{\eta n\varepsilon\}} = \exp\left\{-\eta n\varepsilon + n\sigma^2\eta^2/2\right\},$$

whose minimum is attained at $\eta = \min\left(\frac{\varepsilon}{\sigma^2}, \zeta\right)$. Thus, for any $\varepsilon \leq \sigma^2\zeta$, we have

$$\mathbb{P}\left(Z > n\varepsilon\right) \leq \exp\left\{-\frac{n\varepsilon^2}{2\sigma^2}\right\}.$$

Repeating this argument for $-Z_{st}^{(i)}$ instead of $Z_{st}^{(i)}$, we obtain the same bound for $\mathbb{P}(-Z > n\varepsilon)$. Combining these two bounds yields

$$\mathbb{P}\left(\left|\frac{1}{n}\sum_{i=1}^n \phi(X_s^{(i)}, X_t^{(i)}) - \mathbb{E}_{\theta^*}[\phi(X_s, X_t)]\right| > \varepsilon\right) = \mathbb{P}\left(|Z| > n\varepsilon\right) \leq 2\exp\left\{-\frac{n\varepsilon^2}{2\sigma^2}\right\}.$$

The second inequality can be similarly proved. This completes the proof. $\qquad\square$

Let us define $\gamma_n := \|\nabla L(\theta^*; \mathbb{X}_n)\|_{2,\infty}$. The following lemma indicates that under Assumption 1, with overwhelming probability, $\gamma_n$ approaches zero at the rate of $O(\sqrt{\max\{q, r\}\ln p/n})$.

**Lemma 2.** *Assume that Assumption 1 is valid. If $n > 6\max\{q, r\}\ln p/(\sigma^2\zeta^2)$, then with probability at least $1 - 2\max\{q, r\}p^{-1}$ the following inequality holds:*

$$\gamma_n = \|\nabla L(\theta^*; \mathbb{X}_n)\|_{2,\infty} \leq \sigma\sqrt{6\max\{q, r\}\ln p/n}.$$

*Proof.* From the gradient term (6) and Lemma 1 we have the following inequalities hold for any $\varepsilon < \sigma^2\zeta$:

$$\mathbb{P}\left(\left|\frac{\partial L(\theta^*; \mathbb{X}_n)}{\partial \theta_{s,k}^*}\right| > \varepsilon\right) = \mathbb{P}\left(\left|\frac{1}{n}\sum_{i=1}^n \varphi_k(X_s^{(i)}) - \mathbb{E}_{\theta^*}[\varphi_k(X_s)]\right| > \varepsilon\right) \leq 2\exp\left\{-\frac{n\varepsilon^2}{2\sigma^2}\right\},$$

$$\mathbb{P}\left(\left|\frac{\partial L(\theta^*; \mathbb{X}_n)}{\partial \theta_{st,l}^*}\right| > \varepsilon\right) = \mathbb{P}\left(\left|\frac{1}{n}\sum_{i=1}^n \phi_l(X_s^{(i)}, X_t^{(i)}) - \mathbb{E}_{\theta^*}[\phi_l(X_s, X_t)]\right| > \varepsilon\right) \leq 2\exp\left\{-\frac{n\varepsilon^2}{2\sigma^2}\right\}.$$

Let $\theta_s^* = [\theta_{s,1}^*, ..., \theta_{s,q}^*]$ and $\theta_{st}^* = [\theta_{st,1}^*, ..., \theta_{st,r}^*]$. By the union bound we obtain

$$\mathbb{P}\left(\left\|\frac{\partial L(\theta^*; \mathbb{X}_n)}{\partial \theta_s^*}\right\| > \varepsilon\right) \leq \sum_{k=1}^q \mathbb{P}\left(\left|\frac{\partial L(\theta^*; \mathbb{X}_n)}{\partial \theta_{s,k}^*}\right| > \frac{\varepsilon}{\sqrt{q}}\right) \leq 2q\exp\left\{-\frac{n\varepsilon^2}{2q\sigma^2}\right\},$$

$$\mathbb{P}\left(\left\|\frac{\partial L(\theta^*; \mathbb{X}_n)}{\partial \theta_{st}^*}\right\| > \varepsilon\right) \leq \sum_{l=1}^r \mathbb{P}\left(\left|\frac{\partial L(\theta^*; \mathbb{X}_n)}{\partial \theta_{st,l}^*}\right| > \frac{\varepsilon}{\sqrt{r}}\right) \leq 2r\exp\left\{-\frac{n\varepsilon^2}{2r\sigma^2}\right\}.$$

Therefore,

$$\mathbb{P}(\|\nabla L(\theta^*; \mathbb{X}_n))\|_{2,\infty} > \varepsilon) \leq 2qp\exp\left\{-\frac{n\varepsilon^2}{2q\sigma^2}\right\} + 2r(p^2 - p)\exp\left\{-\frac{n\varepsilon^2}{2r\sigma^2}\right\}$$

$$\leq 2\max\{q, r\}p^2\exp\left\{-\frac{n\varepsilon^2}{2\max\{q, r\}\sigma^2}\right\}.$$

Let us choose $\varepsilon = \sigma\sqrt{6\max\{q,r\}\ln p/n}$. Since $n > 6\max\{q,r\}\ln p/(\sigma^2\zeta^2)$, we have $\varepsilon < \sigma^2\zeta$. Therefore we obtain that with probability at least $1 - 2\max\{q,r\}p^{-1}$,

$$\|\nabla L(\theta^*; \mathbb{X}_n)\|_{2,\infty} \leq \sigma\sqrt{6\max\{q,r\}\ln p/n}.$$

This completes the proof of Lemma 2. $\qquad\square$

The following result further bounds the estimation error of the MLE estimator (7) in terms of $\gamma_n$, $\delta$ and $\beta$.

**Lemma 3.** *Assume that the conditions in Assumption 2 hold true. Assume that $\lambda_n \in [2\gamma_n, c_0\gamma_n]$ for some $c_0 \geq 2$. Define $\gamma = 3c_0\sqrt{\|\theta^*\|_{2,0}}\beta^{-1}\gamma_n$. If $\gamma < \delta$, then we have*

$$\|\hat{\theta}_n - \theta^*\| \leq 3c_0\sqrt{\|\theta^*\|_{2,0}}\beta^{-1}\gamma_n.$$

*Proof.* Let $\Delta\theta = \hat{\theta}_n - \theta^*$ and we define $\Delta\tilde{\theta} = t\Delta\theta$ where we pick $t = 1$ if $\|\Delta\theta\| \leq \delta$ and $t \in (0,1)$ with $\|\Delta\tilde{\theta}\| = \delta$ otherwise. By construction we have $\|\Delta\tilde{\theta}\| \leq r$. We now claim that $\|\Delta\tilde{\theta}_{\bar{S}}\|_{2,1} \leq 3\|\Delta\tilde{\theta}_S\|_{2,1}$. Indeed, since $\theta^*_{\bar{S}} = 0$, we have

$$\|\theta^* + \Delta\tilde{\theta}\|_{2,1} - \|\theta^*\|_{2,1} = \|(\theta^* + \Delta\tilde{\theta})_S\|_{2,1} + \|\Delta\tilde{\theta}_{\bar{S}}\|_{2,1} - \|\theta^*_S\|_{2,1} \geq \|\Delta\tilde{\theta}_{\bar{S}}\|_{2,1} - \|\Delta\tilde{\theta}_S\|_{2,1}. \quad \text{(A.1)}$$

From the convexity of function $L(\theta; \mathbb{X}_n)$ and $\lambda_n \geq 2\gamma_n = 2\|\nabla L(\theta^*; \mathbb{X}_n)\|_{2,\infty}$ we have

$$L(\theta^* + \Delta\tilde{\theta}; \mathbb{X}_n) - L(\theta^*; \mathbb{X}_n) \geq \langle\nabla L(\theta^*; \mathbb{X}_n), \Delta\tilde{\theta}\rangle \geq -\|\nabla L(\theta^*; \mathbb{X}_n)\|_{2,\infty}\|\Delta\tilde{\theta}\|_{2,1} \geq -\frac{\lambda_n}{2}\|\Delta\tilde{\theta}\|_{2,1}. \tag{A.2}$$

Due to the optimality of $\hat{\theta}_n$ and the convexity of $L(\theta; \mathbb{X}_n)$, it holds that

$$L(\theta^* + \Delta\tilde{\theta}; \mathbb{X}_n) + \lambda_n\|\theta^* + \Delta\tilde{\theta}\|_{2,1} \leq L(\theta^*; \mathbb{X}_n) + \lambda_n\|\theta^*\|_{2,1}. \tag{A.3}$$

By combining the proceeding three inequalities (A.1), (A.2) and (A.3), we obtain that

$$
\begin{aligned}
0 &\geq L(\theta^* + \Delta\tilde{\theta}; \mathbb{X}_n) + \lambda_n\|\theta^* + \Delta\tilde{\theta}\|_{2,1} - L(\theta^*; \mathbb{X}_n) - \lambda_n\|\theta^*\|_{2,1} \\
&\geq -\frac{\lambda_n}{2}(\|\Delta\tilde{\theta}_S\|_{2,1} + \|\Delta\tilde{\theta}_{\bar{S}}\|_{2,1}) + \lambda_n(\|\Delta\tilde{\theta}_{\bar{S}}\|_{2,1} - \|\Delta\tilde{\theta}_S\|_{2,1}),
\end{aligned}
$$

which implies $\|\Delta\tilde{\theta}_{\bar{S}}\|_{2,1} \leq 3\|\Delta\tilde{\theta}_S\|_{2,1}$. From second-order Taylor expansion we know that there exists a real number $\xi \in [0,1]$ such that

$$L(\theta^* + \Delta\tilde{\theta}; \mathbb{X}_n) = L(\theta^*; \mathbb{X}_n) + \langle\nabla L(\theta^*; \mathbb{X}_n), \Delta\tilde{\theta}\rangle + \frac{1}{2}\Delta\tilde{\theta}^\top\nabla^2 L(\theta^* + \xi\Delta\tilde{\theta}; \mathbb{X}_n)\Delta\tilde{\theta}.$$

By using Assumption 2 (note that $\|\xi\Delta\tilde{\theta}\| \leq \|\Delta\tilde{\theta}\| \leq r$) and (A.2) we have

$$L(\theta^* + \Delta\tilde{\theta}; \mathbb{X}_n) - L(\theta^*; \mathbb{X}_n) \geq \langle\nabla L(\theta^*; \mathbb{X}_n), \Delta\tilde{\theta}\rangle + \frac{\beta}{2}\|\Delta\tilde{\theta}\|^2 \geq -\frac{\lambda_n}{2}\|\Delta\tilde{\theta}\|_{2,1} + \frac{\beta}{2}\|\Delta\tilde{\theta}\|^2. \quad \text{(A.4)}$$

By combining the inequalities (A.1), (A.3) and (A.4), we obtain

$$
\begin{aligned}
0 &\geq L(\theta^* + \Delta\tilde{\theta}; \mathbb{X}_n) + \lambda_n\|\theta^* + \Delta\tilde{\theta}\|_{2,1} - L(\theta^*; \mathbb{X}_n) - \lambda_n\|\theta^*\|_{2,1} \\
&\geq -\frac{\lambda_n}{2}\|\Delta\tilde{\theta}\|_{2,1} + \frac{\beta}{2}\|\Delta\tilde{\theta}\|^2 + \lambda_n(\|\Delta\tilde{\theta}_{\bar{S}}\|_{2,1} - \|\Delta\tilde{\theta}_S\|_{2,1}) \\
&\geq \frac{\lambda_n}{2}(\|\Delta\tilde{\theta}_{\bar{S}}\|_{2,1} - 3\|\Delta\tilde{\theta}_S\|_{2,1}) + \frac{\beta}{2}\|\Delta\tilde{\theta}\|^2 \\
&\geq -1.5\lambda_n\|\Delta\tilde{\theta}_S\|_{2,1} + \frac{\beta}{2}\|\Delta\tilde{\theta}\|^2 \geq -1.5\lambda_n\sqrt{\|\theta^*\|_{2,0}}\|\Delta\tilde{\theta}\| + \frac{\beta}{2}\|\Delta\tilde{\theta}\|^2,
\end{aligned}
$$

which implies that

$$\|\Delta\tilde{\theta}\| \leq 3\lambda_n\beta^{-1}\sqrt{\|\theta^*\|_{2,0}} \leq 3c_0\sqrt{\|\theta^*\|_{2,0}}\beta^{-1}\gamma_n = \gamma.$$

Since $\gamma < \delta$, we claim that $t = 1$ and thus $\Delta\tilde{\theta} = \Delta\theta$. Indeed, if otherwise $t < 1$, then $\|\Delta\tilde{\theta}\| = \delta > \gamma$ which contradicts the above inequality. This completes the proof. $\qquad\square$

Equipped with Lemma 2 and Lemma 3, we are now in the position to prove Theorem 1.

*Proof of Theorem 1.* By invoking Lemma 2 and the condition $n > 54c_0^2\delta^{-2}\beta^{-2}\sigma^2\|\theta^*\|_{2,0}\ln p)$ we have that with probability at least $1 - 2\max\{q,r\}p^{-1}$,

$$\gamma = 3c_0\sqrt{|E|}\beta^{-1}\gamma_n \leq 3c_0\beta^{-1}\sigma\sqrt{6\max\{q,r\}\|\theta^*\|_{2,0}\ln p/n} < \delta.$$

By applying Lemma 3 we obtain the desired result. □

## A.3 Proof of Theorem 2

To prove the theorem, we will need to study the concentration bound of the random variables defined by

$$\tilde{Z}_{s,k} := \mathbb{E}_{\theta_s^*}[\varphi_k(X_s) \mid X_{\backslash s}] - \mathbb{E}_{\theta^*}[\varphi_k(X_s)], \quad \tilde{Z}_{st,l} := \mathbb{E}_{\theta_s^*}[\phi_l(X_s, X_t) \mid X_{\backslash s}] - \mathbb{E}_{\theta^*}[\phi_l(X_s, X_t)].$$

The following lemma shows that under Assumption 1, $\tilde{Z}_{st}$ have exponential-type moment generating function.

**Lemma 4.** *If Assumption 1 holds, then we have that for any* $(s,k)$, $(s,t,l)$, *and for all* $|\eta| \leq \zeta$,

$$\mathbb{E}[\exp\{\eta\tilde{Z}_{s,k}\}] \leq \exp\{\sigma^2\eta^2/2\}, \quad \mathbb{E}[\exp\{\eta\tilde{Z}_{st,l}\}] \leq \exp\{\sigma^2\eta^2/2\}.$$

*Proof.* We only prove the first inequality as the second one can be very similarly proved. Note that for any $\eta$, $\exp\{\eta x\}$ is convex with respect to $x$. By applying Jensen's inequality we have

$$\exp\{\eta\mathbb{E}_{\theta_s^*}[\varphi_k(X_s) \mid X_{\backslash s}]\} \leq \mathbb{E}_{\theta_s^*}[\exp\{\eta\varphi_k(X_s)\} \mid X_{\backslash s}].$$

By taking the expectation $\mathbb{E}_{\theta_{\backslash s}^*}[\cdot]$ with respect to the marginal distribution of $X_{\backslash s}$, and using the rule of iterated expectation, we obtain

$$\mathbb{E}_{\theta_{\backslash s}^*}\left[\exp\{\eta\mathbb{E}_{\theta_s^*}[\varphi_k(X_s) \mid X_{\backslash s}]\}\right] \leq \mathbb{E}_{\theta_{\backslash s}^*}\left[\mathbb{E}_{\theta_s^*}[\exp\{\eta\varphi_k(X_s)\} \mid X_{\backslash s}]\right] = \mathbb{E}_{\theta^*}[\exp\{\eta\varphi_k(X_s)\}].$$

By using the "law of the unconscious statistician" and the above inequality we obtain

$$\mathbb{E}[\exp\{\eta\tilde{Z}_{s,k}\}] \leq \mathbb{E}[\exp\{\eta Z_{s,k}\}] \leq \exp\{\sigma^2\eta^2/2\},$$

where the last inequality follows from Assumption 1. This completes the proof. □

This lemma shows that the random variables $\{\tilde{Z}_{s,k}, \tilde{Z}_{st,l}\}$ all have the same exponential-type moment generating function as that of $\{Z_{s,k}, Z_{st,l}\}$ investigated in the previous subsection.

Based on Lemma 4 and the proof of Lemma 1, we may establish the following lemma which will be used in the proofs to follow.

**Lemma 5.** *If Assumption 1 holds, then for all* $(s,t)$, $(s,t,l)$ *and any* $\varepsilon \leq \sigma^2\zeta$ *we have*

$$\mathbb{P}\left(\left|\frac{1}{n}\sum_{i=1}^n \mathbb{E}_{\theta_s^*}[\varphi_k(X_s) \mid X_{\backslash s}^{(i)}] - \mathbb{E}_{\theta^*}[\varphi_k(X_s)]\right| > \varepsilon\right) \leq 2\exp\left\{-\frac{n\varepsilon^2}{2\sigma^2}\right\},$$

$$\mathbb{P}\left(\left|\frac{1}{n}\sum_{i=1}^n \mathbb{E}_{\theta_s^*}[\phi_l(X_s, X_t^{(i)}) \mid X_{\backslash s}^{(i)}] - \mathbb{E}_{\theta^*}[\phi_l(X_s, X_t)]\right| > \varepsilon\right) \leq 2\exp\left\{-\frac{n\varepsilon^2}{2\sigma^2}\right\}.$$

Let us define $\tilde{\gamma}_n := \|\nabla\tilde{L}(\theta_s^*; \mathbb{X}_n)\|_{2,\infty}$. The following lemma further indicates that under Assumption 1, with overwhelming probability, $\tilde{\gamma}_n$ approaches zero at the rate of $O(\sqrt{\max\{q,r\}\ln p/n})$.

**Lemma 6.** *Assume that Assumption 1 holds. If* $n > 6\max\{q,r\}\ln p/(\sigma^2\zeta^2)$, *then with probability at least* $1 - 4\max\{q,r\}p^{-2}$ *the following inequality holds:*

$$\tilde{\gamma}_n \leq 2\sigma\sqrt{6\max\{q,r\}\ln p/n}.$$

*Proof.* Recall the formulation of gradient $\nabla \tilde{L}(\theta_{\mathbf{s}}; \mathbb{X}_n)$ in (9). For any node $t \in V \setminus s$, we have

$$\left| \frac{\partial \tilde{L}(\theta_{\mathbf{s}}^*; \mathbb{X}_n)}{\partial \theta_{st,l}^*} \right|$$

$$= \left| \frac{1}{n} \sum_{i=1}^n -\phi_l(X_s^{(i)}, X_t^{(i)}) + \mathbb{E}_{\theta_{\mathbf{s}}^*}[\phi_l(X_s, X_t^{(i)}) \mid X_{\setminus s}^{(i)}] \right|$$

$$\leq \left| \frac{1}{n} \sum_{i=1}^n \phi_l(X_s^{(i)}, X_t^{(i)}) - \mathbb{E}_{\theta^*}[\phi_l(X_s, X_t)] \right| + \left| \frac{1}{n} \sum_{i=1}^n \mathbb{E}_{\theta_{\mathbf{s}}^*}[\phi_l(X_s, X_t^{(i)}) \mid X_{\setminus s}^{(i)}] - \mathbb{E}_{\theta^*}[\phi_l(X_s, X_t)] \right|.$$

Therefore, for any $\varepsilon \leq 2\sigma^2 \zeta$,

$$\mathbb{P}\left( \left| \frac{\partial \tilde{L}(\theta_{\mathbf{s}}^*; \mathbb{X}_n)}{\partial \theta_{st,l}^*} \right| > \varepsilon \right) \leq \mathbb{P}\left( \left| \frac{1}{n} \sum_{i=1}^n \phi_l(X_s^{(i)}, X_t^{(i)}) - \mathbb{E}_{\theta^*}[\phi_l(X_s, X_t)] \right| > \frac{\varepsilon}{2} \right)$$

$$+ \mathbb{P}\left( \left| \frac{1}{n} \sum_{i=1}^n \mathbb{E}_{\theta_{\mathbf{s}}^*}[\phi_l(X_s, X_t^{(i)}) \mid X_{\setminus s}^{(i)}] - \mathbb{E}_{\theta^*}[\phi_l(X_s, X_t)] \right| > \frac{\varepsilon}{2} \right)$$

$$\overset{\xi_1}{\leq} 4 \exp\left\{ -\frac{n\varepsilon^2}{8\sigma^2} \right\},$$

where $\xi_1$ follows from Lemma 1 and Lemma 5. Similarly, we can show

$$\mathbb{P}\left( \left| \frac{\partial \tilde{L}(\theta_{\mathbf{s}}^*; \mathbb{X}_n)}{\partial \theta_{s,k}^*} \right| > \varepsilon \right) \leq 4 \exp\left\{ -\frac{n\varepsilon^2}{8\sigma^2} \right\}.$$

By the union bound we obtain

$$\mathbb{P}\left( \left\| \frac{\partial \tilde{L}(\theta_{\mathbf{s}}^*; \mathbb{X}_n)}{\partial \theta_s^*} \right\| > \varepsilon \right) \leq \sum_{k=1}^q \mathbb{P}\left( \left| \frac{\partial \tilde{L}(\theta_{\mathbf{s}}^*; \mathbb{X}_n)}{\partial \theta_{s,k}^*} \right| > \frac{\varepsilon}{\sqrt{q}} \right) \leq 4q \exp\left\{ -\frac{n\varepsilon^2}{8q\sigma^2} \right\},$$

$$\mathbb{P}\left( \left\| \frac{\partial \tilde{L}(\theta_{\mathbf{s}}^*; \mathbb{X}_n)}{\partial \theta_{st}^*} \right\| > \varepsilon \right) \leq \sum_{l=1}^r \mathbb{P}\left( \left| \frac{\partial \tilde{L}(\theta_{\mathbf{s}}^*; \mathbb{X}_n)}{\partial \theta_{st,l}^*} \right| > \frac{\varepsilon}{\sqrt{r}} \right) \leq 4r \exp\left\{ -\frac{n\varepsilon^2}{8r\sigma^2} \right\}.$$

This implies

$$\mathbb{P}(\|\nabla \tilde{L}(\theta_{\mathbf{s}}^*; \mathbb{X}_n)\|_{2,\infty} > \varepsilon) \leq 4q \exp\left\{ -\frac{n\varepsilon^2}{8q\sigma^2} \right\} + 4r(p-1) \exp\left\{ -\frac{n\varepsilon^2}{8r\sigma^2} \right\} \leq 4 \max\{q,r\} p \exp\left\{ -\frac{n\varepsilon^2}{8\max\{q,r\}\sigma^2} \right\}.$$

Let us choose $\varepsilon = 2\sigma\sqrt{6\max\{q,r\}\ln p/n}$. Since $n > 6\max\{q,r\}\ln p/(\sigma^2\zeta^2)$, we have $\varepsilon < 2\sigma^2\zeta$. We conclude that with probability at least $1 - 4\max\{q,r\}p^{-2}$,

$$\|\nabla \tilde{L}(\theta_{\mathbf{s}}^*; \mathbb{X}_n)\|_{2,\infty} \leq 2\sigma\sqrt{6\max\{q,r\}\ln p/n}.$$

This proves the desired bound. $\qquad\square$

The following result establishes the estimation error of the node-conditional estimator (10) in terms of $\tilde{\gamma}_n$, $\tilde{\delta}$ and $\tilde{\beta}$.

**Lemma 7.** *Assume that the conditions in Assumption 3 hold. Assume that $\lambda_n \in [2\tilde{\gamma}_n, \tilde{c}_0\tilde{\gamma}_n]$ for some $\tilde{c}_0 \geq 2$. Define $\tilde{\gamma} = 3\tilde{c}_0\sqrt{\max\{q,r\}(d+1)}\tilde{\beta}^{-1}\tilde{\gamma}_n$. If $\tilde{\gamma} < \tilde{\delta}$, then we have*

$$\|\hat{\theta}_{\mathbf{s}}^n - \theta_{\mathbf{s}}^*\| \leq 3\tilde{c}_0\sqrt{\max\{q,r\}(d+1)}\tilde{\beta}^{-1}\tilde{\gamma}_n.$$

The proof of this lemma mirrors that of Lemma 3.

Based on the above lemmas, we can now complete the proof of Theorem 2.

*Proof of Theorem 2.* From Lemma 6 and the condition $n > 216\tilde{c}_0^2\tilde{\delta}^{-2}\tilde{\beta}^{-2}\sigma^2\max\{q,r\}\|\theta_{\mathbf{s}}^*\|_{2,0}\ln p$ we know that with probability at least $1 - 4\max\{q,r\}p^{-2}$,

$$\tilde{\gamma} = 3\tilde{c}_0\sqrt{\|\theta_{\mathbf{s}}^*\|_{2,0}}\tilde{\beta}^{-1}\tilde{\gamma}_n \leq 6\tilde{c}_0\tilde{\beta}^{-1}\sigma\sqrt{6\|\theta_{\mathbf{s}}^*\|_{2,0}\ln p/n} < \tilde{\delta}.$$

By applying Lemma 7 we obtain the desired result. $\qquad\square$

# B  Some Additional Remarks on Assumptions

We provide here a few additional remarks on the conditions under which Assumption 1 and Assumption 3 can be satisfied.

**Remark 4** (On Assumption 1: the basis $\{\varphi_k(X_s), \phi_l(X_s, X_t)\}$ are bounded). *It can be verified that Assumption 1 holds when the basis $\{\varphi_k(X_s), \phi_l(X_s, X_t)\}$ are bounded. Indeed, given the bounded basis, $\{Z_{s,k}, Z_{st,l}\}$ are bounded random variables with zero means. Based on the Hoeffding's Lemma, for any random variable $Z \in [a, b]$ and $\mathbb{E}[Z] = 0$, we have $\mathbb{E}[\exp\{\eta Z\}] \leq \exp\{\eta^2(b-a)^2/8\}$ holds for all scalar $\eta$. Therefore Assumption 1 holds when the basis statistics $\{\varphi_k(X_s), \phi_l(X_s, X_t)\}$ are bounded.*

**Remark 5** (On Assumption 1: the basis $\{\varphi_k(X_s), \phi_l(X_s, X_t)\}$ are unbounded but sub-exponential). *We call a random variable $Z$ sub-exponential if there exist constants $c_1, c_2 > 0$ such that $\mathbb{P}(|Z - \mathbb{E}(Z)| > \eta) \leq \exp\{c_1 - \eta/c_2\}$, for all $\eta > 0$. Using the result in [23, Lemma 5.15], we can verify that Assumption 1 holds when $\{\varphi_k(X_s), \phi_l(X_s, X_t)\}$ are sub-exponential random variables. For instance, consider that the energy function in (4) satisfies $B(X; \theta^*) \leq -\frac{1}{2}(X - \mu)^\top \Omega (X - \mu) + c$ for some constant vector $\mu$, scalar $c$ and positive-definite matrix $\Omega \succ 0$. It can be verified that the marginal distribution of $X_s$ is bounded by $\mathbb{P}(X_s) \leq c_s \exp\left\{-\frac{(X_s - \mu_s)^2}{2\sigma_s^2}\right\}$ for some constants $c_s, \mu_s$ and $\sigma_s$. If further assuming there exist constants $c_k > 0$ and $c'_k$ such that $\varphi_k(X_s) \leq c_k(X_s - \mu_s)^2 + c'_k$, then we claim that $Z_{s,k}$ is sub-exponential. Indeed, for any $\eta > 0$, by using Markov inequality we have*

$$
\begin{aligned}
\mathbb{P}(\varphi_k(X_s) > \eta) &\leq \frac{\mathbb{E}[\exp\{\varphi_k(X_s)/(4c_k\sigma_s^2)\}]}{\exp\{\eta/(4c_k\sigma_s^2)\}} \\
&\leq \frac{c_s \int_{\mathcal{X}} \exp\{-(X_s - \mu_s)^2/(4\sigma_s^2)\} dX_s}{\exp\{(\eta - c'_k)/(4c_k\sigma_s^2)\}} \\
&\propto \exp\{-(\eta - c'_k)/(4c_k\sigma_s^2)\},
\end{aligned}
$$

*which shows that $\varphi_k(X_s)$ is a sub-exponential random variable and so is $Z_{s,k}$. Similarly, we can show that $\phi_l(X_s, X_t)$ is sub-exponential if there exist $c_l > 0$ and $c'_l$ such that $\phi_l(X_s, X_t) \leq c_l((X_s - \mu_s)^2 + (X_t - \mu_t)^2) + c'_l$. Clearly, the analysis made for this example is applicable to the multivariate Gaussian for which some similar results have been established in [17, 16].*

**Remark 6** (On Assumption 3). *Assumption 3 requires that the Hessian $\nabla^2 \tilde{L}(\theta_\mathbf{s}; \mathbb{X}_n)$ is positive definite in the cone $\tilde{\mathcal{C}}_S$ when $\theta_\mathbf{s}$ lies in a local ball centered at $\theta_\mathbf{s}^*$. Specially, when $X$ is multivariate Gaussian, i.e., $\phi(X_s, X_t) = X_s X_t$ and $f(X_s) = -X_s^2$, this condition essentially requires that the design matrix $A_s^n = \frac{1}{n}\sum_{i=1}^n X_{\backslash s}^{(i)}(X_{\backslash s}^{(i)})^\top$ is positive definite. In this case, if the precision matrix is positive definite, then it is known from the compressed sensing literature [3, 4] that with overwhelming probability, $A_s^n$ is positive definite provided that the sample size $n = O(\ln p)$ is sufficiently large. More generally, it can be verified that $\mathbb{E}[\nabla^2 \tilde{L}(\theta_\mathbf{s}; \mathbb{X}_n)]$ is the sub-matrix of $\nabla^2 L(\theta; \mathbb{X}_n)$ associated with the pairs $\{(s, t) \mid t \in V \setminus \{s\}\}$. Therefore, if the whole Hessian matrix $\nabla^2 L(\theta; \mathbb{X}_n)$ is positive definite at any $\theta$, then $\mathbb{E}[\nabla^2 \tilde{L}(\theta_\mathbf{s}; \mathbb{X}_n)]$ is also positive definite. By using weak law of large number we get that Assumption 3 holds with high probability when $n$ is sufficiently large.*