[Reviews · NeurIPS 2016]

Reviewer 1

Summary

This paper discusses the problem of fitting a parametric exponential family graphical model with some consistency results presented.

Qualitative Assessment

First, the proposed method is parametric but instead of semi parametric. Second, the assumption of this paper is rather strong and hard to verify.

Confidence in this Review

2-Confident (read it all; understood it all reasonably well)


Reviewer 2

Summary

The paper proposes a procedure for estimating a structure of a subclass of exponential family graphical models and shows that under appropriate conditions the estimation procedures succeeds with high probability.

Qualitative Assessment

Learning graphical models is an important problem. While this paper contains some interesting ideas, theoretical properties established and numerical experiments performed are lacking as I explain below. On the theoretical side, main theorems follow the strategy presented in Negahban et al paper on M-estimation. The new part here is the results that allows for estimation even when the normalizing constant is approximated via Monte-Carlo. An unsatisfactory part of the theory is that the assumptions 1-3 are simply assumed. Remarks 4-6 (in appendix) comment on the assumptions in some special cases. However, unlike the work of Yang et al (JMLR 2015), the assumptions are verified for a small subset of model classes. For example, Remark 6 only considers parameteric Gaussian model. The paper "Estimation of high-dimensional graphical models using regularized score matching" by Lina Lin, Mathias Drton, and Ali Shojaie in EJS essentially studies estimation in the same model family. That paper uses score matching objective to learn parameters. These two procedures should be compared both numerically and theoretical conditions should be compared. Furthermore, the procedure by Lin et al does not require numerical integration or Monte-Carlo approximation, so potentially could be faster and more precise. The paper "On Semiparametric Exponential Family Graphical Models" by Zhuoran Yang, Yang Ning, Han Liu at https://arxiv.org/abs/1412.8697 studies semiparametric models as well. How does procedure described here compare to the one by Yang et al -- both theoretically and numerically? As a side note, the model considered here is not semiparametric, as the knowledge of function f_s and f_{s,t} is assumed to be known exactly. What happens if the expansion in terms of basis is only approximate?

Confidence in this Review

3-Expert (read the paper in detail, know the area, quite certain of my opinion)


Reviewer 3

Summary

This paper is interested with semi-parametric exponential family graphical models, in the case when all concerned functions belong to a parametric set defined by some basis functions. In this case, the problem is equivalent to estimating a high-dimensional vector, but the authors show that if the graph is sparse, the model can be estimated more efficiently, using a group-lasso-like estimator. Then, they show how to compute their estimator in practice and display experimental results.

Qualitative Assessment

This paper tackles an interesting and important problem. The results extend well-known ideas in classical regression to a more challenging model, if I am not mistaking, especially results from ORACLE INEQUALITIES AND OPTIMAL INFERENCE UNDER GROUP SPARSITY by Lounici and colleagues. Actually, I am quite surprised that you do not cite Simultaneous analysis of lasso and Dantzig selector by Bickel and colleagues, since you use a kind of restricted eigenvalue condition that was introduced in this paper. On the other hand, I think that the assumption (3) is rather strong, but maybe you could write simple conditions on the norm of the difference between both sides of the inequalities (3) that allow to get the same results?

Confidence in this Review

2-Confident (read it all; understood it all reasonably well)


Reviewer 4

Summary

This paper proposes parameter estimators for exponential family graphical models in which the univariate and pairwise sufficient statistics are representable as linear combinations of pre-defined basis functions. Two regularized maximum likelihood estimators are presented, the first for joint estimation of all parameters, the second for conditional estimation of the parameters associated with each node. A Monte-Carlo approximation to certain expectations is used to simplify the algorithms. Finite-sample statistical guarantees are established on the estimation error, which agree with existing results on Gaussian and nonparanormal graphical models. Experimental results show greater flexibility in modeling nonlinear interactions between variables. I think this last aspect is the main strength of the paper compared to existing approaches, in addition to the theoretical results.

Qualitative Assessment

Most important: 1. In Theorem 1, the error rate is as expected since max{q,r} ||theta*||_{2,0} is approximately the number of parameters in the model. I think the factor of max{q,r} should not be omitted from Section 1.1, last paragraph so as not to be misleading. I am surprised however that the same factor of max{q,r} is not present in the error rate in Theorem 2, being replaced instead by (q+r) inside the logarithm. This result should be checked since if I understand correctly, the number of parameters is still something like q or r times ||theta*_s||_{2,0}. If Theorem 2 is correct as currently stated then a more thorough justification in words is needed. Major comments: 2. I think it may be inaccurate to call the model "semi-parametric" because the set of basis functions is finite and fixed. Moreover the dimensions of the bases used in the experiments are small (< 10). 3. Related to the point above, I believe the proposed approach shifts the burden onto choosing an appropriate set of basis functions for the data at hand. For the simulated data in Section 4.1, the bases are chosen to be a perfect fit or at least a good fit to the true form of the interactions. A case where the basis is not well-matched would be of interest. In general, the choice of basis is an issue deserving of more attention. 4. The definition of groups is not clear in the Notations subsection and remains so until below (7) where the (2,1)-norm is defined. Why are node groups penalized the same as edge groups? 5. In Proposition 1, definition of sigmahat, should it be sigmahat^2 instead? The RHS looks like a variance. 6. I am not seeing how the number of Monte Carlo samples m was chosen in the experiments. 7. In Section 4.2, the use of stock category as ground truth for links seems a bit questionable since stocks in different categories could also be strongly dependent, while not all stocks in the same category may be dependent. Minor issues: 8. Section 1, paragraph 1: "undirected graph consists" --> "undirected graph consisting" 9. In Section 1.2, the second half of the paragraph could be better organized. Do [9,27] belong to the nonparanormal approach? It appears that [7,20,8] are fully non-parametric methods. This could be emphasized with a brief discussion of the advantages/disadvantages of non-parametric models vs. semi-parametric ones and the proposed model. 10. Below (10) in the definition of ||theta||_{2,1}, I think the summation should be over t in N(s). 11. Just above Proposition 1, is "A.1" an appendix in the supplementary material? 12. In Theorems 1 and 2, the constants c_0 and ctilde_0 are undefined. In Theorem 2, there may be missing tildes e.g. on delta. 13. In Section 4.1, is performance reported on the training sample, after the tuning sample has been used to select lambda_n?

Confidence in this Review

3-Expert (read the paper in detail, know the area, quite certain of my opinion)


Reviewer 5

Summary

They consider the problem of estimation of the sparse parameters of the additive exponential family graphical models using two estimators and they also provide statistical guarantees for those two methods that are similar to the existing results in literature order-wise.

Qualitative Assessment

1. More explanation about the assumptions would be nice. Does relevant literature (like [15,17,9,27]) utilize some similar kind of assumption? 2. They provide two optimization problems for estimating desired parameters of AdEFGM with statistical guarantees. They also suggest a proximal gradient method to solve those optimizations. Their model is more general compared to the one in the references and the paper is also self contained in the sense that it provides a method, algorithm and also guarantee for the problem.

Confidence in this Review

2-Confident (read it all; understood it all reasonably well)